# The Impact of Crime against a Person on Domestic Investment in Dubai

**Hatem Adela [1,2,*] and Wadeema Aldhaheri [3]**

[1] Department of Economics, American University in the Emirates, Dubai P.O. Box 28282, United Arab Emirates
[2] Department of Economics, Sadat Academy for Management Sciences, Cairo 11837, Egypt
[3] College of Islamic Studies, Mohamed Bin Zayed University for Humanities, Signal 23, Abu Dhabi P.O. Box 106621, United Arab Emirates; wadeema.aldhaheri@mbzuh.ac.ae
[*] Correspondence: hatem.adileh@aue.ae

**Abstract:** The rise in crime against a person in rapidly growing cities poses significant risks to societies and economies, affecting both microeconomic and macroeconomic aspects. This trend could potentially deter economic performance and domestic investment. Consequently, this study aims to analyze the impact of crime against a person on domestic investment in Dubai spanning 1989–2021. Dubai is considered an emerging economy and a highly competitive global city. It is also acknowledged as one of the world's smart cities. This study employed the novel nonlinear autoregressive distributed lag (NARDL) approach to investigate the impact of both the escalation and contraction of crime against a person on domestic investment in Dubai. The findings exhibit that the fluctuation in crime against a person has an asymmetrical impact on domestic investment. In addition, estimations of the positive and negative long-run asymmetric coefficients indicate that crime against a person has a negative impact on domestic investment in Dubai.

**Keywords:** domestic investment; crime against a person; gross domestic product; nonlinear autoregressive distributed lag; long-run relationship; Dubai

## 1. Introduction

The economic analysis of crime first appeared in the literature within the formulation of an economic model by Becker (1968), who presented crime as an important economic activity (Becker 1968). It involves significant threats to societies and economies at both the microeconomic and macroeconomic levels. Subsequent studies focused on the socioeconomic factors impacting criminal behavior, the unequal allocation of resources, and the processes of socioeconomic dynamics that contribute to poverty. In this framework, various variables are considered, including rapid and uncontrolled urbanization, income and social inequality, poverty, youth unemployment, and deficiencies in the justice system. These factors contribute to increasing economic and political vulnerability. These variables have previously been modeled and analyzed to understand their influence on crime rates, utilizing empirical analyses to investigate whether economic challenges lead to higher crime rates (e.g., Mohammadi et al. 2022; Nguyen et al. 2023; LaFree and Jiang 2023; Goulas and Zervoyianni 2015; Matteo and Petrunia 2021; Montolio 2018; Cornwell and Trumbull 1994; Okpokwasili 2015; Detotto and Pulina 2013). These studies have led to policy suggestions that aid in avoiding and addressing social unrest and criminality, such as boosting people's income, educational attainment, and declining economic problems, particularly inequality and unemployment. Also, crimes against a person impair the accumulation of human capital, disrupt an investment environment that is conducive for both local and international investors, and negatively affect the tourism industry. Consequently, it is considered a factor that determinately influences economic growth and domestic investment across various channels (Giusti and Raya 2019; Goulas and Zervoyianni 2015; Biagi and Detotto 2014; Neanidis and Papadopoulou 2013).

Another aspect of investigating crime from an economic perspective is assessing its detrimental impact on economic growth. In this regard, addressing concerns related to justice and security becomes pivotal for a nation's economic performance (Acemoglu and Johnson 2005).

Therefore, a rise in the crime rate in a society can lead to political instability. Crimes are perceived as a significant hindrance that deters advancement in achieving sustainable development goals (OECD 2016).

Crime against a person includes a broad spectrum of criminal behaviors, typically characterized by causing physical harm, the potential for physical harm, or actions carried out against an individual's will. These offenses encompass acts such as homicide, harassment, stalking, kidnapping, assault, and domestic violence, all of which involve bodily harm or the imminent threat of bodily harm (Alabi and Balogun 2023). Therefore, it could significantly impact the economy, especially domestic and foreign investments, because investors' expectations are largely shaped by security, which is considered the primary factor influencing their investment decisions.

In recent years, Dubai has garnered attention for its vigorous attempts to reduce crime rates, driven by a strong ambition to achieve economic sustainability and enhance domestic investment. The strategy of the Dubai Police is based on preventing and reducing crime by implementing several crime-fighting initiatives that harness the power of artificial intelligence and advanced technology to bolster public safety and security. These initiatives encompass diverse programs, such as Dubai Police's Smart Homes Transformation Plan 2018–2021 (Dubai Police, Open Data 2023), which aimed to entirely convert residences in the Emirate of Dubai into smart homes, enabling homeowners to monitor their homes remotely with a 100% success rate. The International Crime-Fighting Program is designed to adopt a professional approach to tracking criminal activities worldwide, strengthening partnerships in crime detection, and apprehending international criminal organizations. The Criminal Monitoring Program aims to control the behavior of convicted criminals upon their release from prison, ensuring their compliance with court orders and supporting their rehabilitation efforts. The Criminal Behavior Analysis Program aims to analyze and preserve criminal methods, linking their methodologies to their perpetrators. The Alert Program aims to scrutinize jurisdictional areas, aligning them to crime indicators and promptly issuing alerts for potential crimes in specific regions by categorizing them based on their level of concern. The Criminal Information Program catalogs crimes, identifying the elements and parties involved and documenting relevant information to aid crime prevention efforts. The Eyes System, which was launched by Dubai Police in 2017, effectively controls all government-owned surveillance cameras using artificial intelligence (Dubai Police, Open Data 2023). Additionally, it monitors suspects, enabling preemptive action before criminal activities are conducted. The Evidence Seizure System, regarded as one of the most accurate and systematic in the world as of 2022, ensures a 99.2% success rate in uncovering crimes through the examination and analysis of criminal evidence by Dubai Police. The Security Map Program identifies high-crime areas and documents them based on the geographical locations of criminal incidents (Dubai Police, Open Data 2023). Dubai aspires to be the world's foremost tourist destination and a leading investment and financial hub in the Middle East. Therefore, it is committed to adopting cutting-edge technology to ensure the highest safety standards, with a specific emphasis on reducing crime against a person. In 2021 and 2022, Dubai secured an impressive fourth-place ranking in the Global Law and Order Index and Night Traffic Index, outperforming countries like Switzerland, Saudi Arabia, Egypt, and Denmark (Gallup Global Law, and Order 2022).

Therefore, this study focuses on investigating the influence of crime against a person on domestic investment in a burgeoning metropolis like Dubai. Dubai offers a unique case for exploring the relationship between crime and its repercussions on investment. Dubai is widely recognized as a dynamic commercial center (Breslow 2020) and a leading global tourist hotspot. The city underscores the importance of attracting foreign investment and promoting the expansion of domestic investment as essential components of its economic

development. Furthermore, Dubai hosts a diverse community of expatriates from more than 200 countries. It also deviates from typical socioeconomic drivers of criminal behaviors, such as unemployment or poverty. The city prides itself on a high standard of living, characterized by a minimal unemployment rate and a lack of impoverished segments within its population. Consequently, the primary trigger for violent inclinations and criminal behavior is often the perception of insecurity (OECD 2016).

The study consists of five sections, as follows. The first is the introduction, followed by the literature review. The third section describes the data and the descriptive analysis. The fourth section presents the methodology. The fifth section exhibits the empirical results and discussion, while the concluding section encompasses conclusions and policy implications.

## 2. Literature Review

Numerous economic theories have sought to explain how criminal activity impacts an economy. Most of these explanations are based on the principles of rationality (Nguyen et al. 2023). Among these studies, a prominent investigation was conducted by Becker (1968), indicating that in the realm of criminal activity, criminals view crime as an economic venture that aligns with their motivations and incentives, akin to any other economic pursuit (Street 2019). They strategically select targets that promise the greatest rewards while demanding minimal effort and risk. Costs include not only overt opportunity costs, such as sacrificing legitimate employment, but also concealed and unpredictable expenses stemming from potential imprisonment, societal dislocation, and the psychological strain of engaging in illicit acts. The opportunity cost for Dubai's economy represents the direct costs stemming from the loss of production elements, such as cases of homicide resulting in a decline in gross domestic product, costs associated with preventing crime and rehabilitation of criminals, the cost of restrictions on human activities, and the diversion of economic resources and economic activity. In addition, there are indirect costs related to the negative impacts on gross domestic product due to multiplier effects, the cost of medical care, diminishing investment in human capital, lower domestic and foreign investment due to uncertainty and investment costs, and the tendency of small businesses to move towards the informal economy. Therefore, the opportunity cost for society can be calculated by the difference between the level of GDP in the absence of crime and the value of GDP in the event of crime (Trisnawati et al. 2019).

Empirical studies subsequent to Becker's study offer diverse perspectives on the effect of crime on the economy. While most of these studies found a significant negative impact of crime on economic growth, some indicated a positive impact of specific crimes. Other studies found an absence of an impact on economic growth. Raj and Kalluru (2023) explored the effect of homicide on economic growth in India spanning the period 1990–2019 utilizing the autoregressive distributed lag (ARDL) model. The findings indicated that the homicide rate had a significant negative impact on economic growth in the short run. A rise in the homicide rate of 1% led to a 0.25% reduction in economic growth. Lacoe et al. (2018) explored the relationship between crime and private investment shaping the development of urban neighborhoods in Chicago and Los Angeles from 2006 to 2011 by employing OLS, negative binomial, and Poisson models. The results revealed that investors exhibited different behaviors in scenarios of rising crime rates compared to declining crime rates. In particular, the influence of decreasing crime rates on building permits was notably less significant than the impact of increasing crime rates on private investment. Schwartz et al. (2022) investigated the causal effect of recurring exposure to violent crime on test scores in New York City using value-added models and a regression discontinuity approach. The results revealed that violent crime had a negative effect on academic performance, potentially influencing long-term economic growth. Poveda et al. (2019) examined the relationship between homicides, economic growth, and corruption in Colombia over the period 2001–2015 by employing the generalized method of moments (GMM) with panel data. The results indicated that corruption and homicide negatively impacted business development and economic growth. Anser et al. (2020) examined the

relationship between the growth–inequality–poverty (GIP) triangle and crime rate within the framework of the Kuznets' inverted U curve and pro-poor growth scenario using panel data from 16 countries spanning the period 1990–2014 and employing the generalized method of moments (GMM). The findings revealed a linear relationship between per capita income and crime rates as well as a U-shaped relationship between poverty and per capita income. Additionally, an inverted U-shaped relationship was observed between income inequality and economic growth in selected countries. Rueda and Perez (2015) investigated the impact of crime on private investment across 11 South American countries by employing an autoregressive model with a panel spanning from 2000 to 2010. The results suggested that crime negatively impacted private investment. Detotto and Pulina (2013) examined the impact of economic variables and deterrent variables on criminal behavior and identified the types of crime that adversely affected economic activity in Italy from 1970 to 2004 using the autoregressive distributed lag (ARDL) model and the Granger causality test. The results indicated that a lack of deterrence measures positively correlated with crime rates. Moreover, all forms of crime decreased economic activity by reducing the employment rate. Additionally, homicides, extortion robberies, and kidnappings had a crowding-out effect on economic growth.

Lobonţ et al. (2017) examined the relationship between crime and socioeconomic factors in Romania from 1990 to 2014 by employing cointegration analysis and the Granger causality test. The results revealed a significant causal relationship between socioeconomic factors and crime rates. Furthermore, the rise in income inequality had a particularly positive effect on crime rates. Additionally, the location of residence played a crucial role, with urban agglomerations significantly influencing crime rates. Ulucak et al. (2020) examined the socioeconomic factors of crime and the consequences of crime on economic growth using panel data techniques across 25 European countries, including Turkey, from 1993 to 2012. The findings indicated that crime had a negative impact on the economy. Also, an increase in economic variables such as welfare level and income was associated with a decrease in criminal activities, whereas unemployment, inequality, and price level led to an increase in criminal activities. Ajide (2021) investigated the relationship between remittance receipts and crime control in Nigeria from 1986 to 2017 utilizing dynamic ordinary least squares (DOLS), vector autoregression impulse response, the Toda and Yamamoto causality approach, and variance decomposition. The results suggested that remittance receipts had a negative impact on the crime rate. Moreover, a positive shock to remittance inflows resulted in a reduced crime level. Additionally, the Toda and Yamamoto causality test revealed a unidirectional causality, with remittance inflow having an impact on criminal activities in Nigeria. Further studies have shown that some types of crime can positively influence economic growth. Mulok et al. (2017) investigated the relationship between crime and economic growth in Malaysia spanning from 1980 to 2013 using the autoregressive distributed lag (ARDL) model to estimate the long-run relationship and the direction of causation between crime and economic growth. The results showed strong evidence of long-run cointegration, and there was a positive impact of economic growth on crime. Also, in the short run, there was a significant bidirectional correlation between crime and economic growth. Trabelsi and Trabelsi (2021) examined the relationship between corruption and economic growth using a panel data analysis for 88 countries over the period 1984–2011 and a cross-sectional framework in which the growth rate and the International Country Risk Guide (ICRG) index were observed only once for each country. The findings suggested that both high and low corruption levels could decrease economic growth, whereas a moderate level of corruption could have advantages for economic growth if the marginal benefits of corruption are equal to the marginal costs.

On the other hand, some studies have identified situations where there is no discernible impact of crime on economic growth. Mohammed et al. (2022) investigated the relationship between organized crime, corruption, and economic growth for the Economic Community of West African States by analyzing data from 11 countries in the ECOWAS region and employing both the fixed effects model (FEM) and the feasible generalized least

squares (FGLS) method. The study concluded that organized crime did not significantly impact economic growth, but corruption played a substantial role in reducing economic growth within the ECOWAS region. Ashby and Ramos (2013) investigated the relationship between organized crime and foreign direct investment for some industries in Mexico, using homicide as a proxy for the presence of organized crime during the period 2004–2010. The study findings indicated that organized crime did not have a significant impact on foreign investment in the oil and mining sectors. However, it had a detrimental effect on foreign direct investment in the financial services, commerce, and agriculture sectors.

Therefore, this paper contributes to the literature by investigating the relationship between crime against a person and domestic investment in Dubai, which is an emerging economy that stands as a highly competitive global city and is recognized as one of the world's smart cities, ranking 17th on the IMD Smart City Index in 2023 (IMD 2023). Furthermore, it is characterized by a diverse population with over 200 nationalities, who have different ways of obtaining benefits through legal or illegal conduct. This disparity could have a significant impact on the rates of crime against individuals, which in turn is reflected in domestic investment, a pivotal aspect that Dubai prioritizes with utmost importance. In addition, the study sheds light on the significance of such research in major competitive cities with large expatriate populations.

## 3. Data Description and Descriptive Analysis

In this study, annual time series data spanning the period 1989–2021 were used. The variables invl, gdpl, tral, and crpl were used, which are the primary explanatory factors that wield significant influence on domestic investment in Dubai, particularly gross domestic product (GDP) and UAE trade openness. This is demonstrated by data on merchandise exports and imports, which show Dubai's nonoil foreign trade constitutes about 75% of the UAE's total nonoil trade (https://www.moec.gov.ae/en/open-data, accessed on 5 July 2023) The definitions and sources of the variables are reported in Table 1. The variable data were transformed into logarithms to address the heteroskedasticity problem (Khan et al. 2019).

**Table 1.** Definition and sources of variables.

| Variable | Definition | Sources |
|---|---|---|
| invl: | domestic investment (gross fixed capital formation by economic activity), | Dubai Statistics Center https://www.dsc.gov.ae/ar-ae/Themes/Pages/National-Accounts.aspx?Theme=24 (accessed on 7 July 2023) |
| gdpl | real gross domestic product (100 = 2010) in local currency, | Dubai Statistics Center https://www.dsc.gov.ae/ar-ae/Themes/Pages/National-Accounts.aspx?Theme=24 (accessed on 7 July 2023) |
| tral | Trade openness (total trade of merchandise of Exports & Imports) | UNCTAD https://unctadstat.unctad.org/wds/TableViewer/tableView.aspx (accessed on 7 July 2023) |
| crpl | Crime against a person÷ | Dubai Police General Head Quarters, & Public Prosecution (Publish and Unpublished data) https://www.dubaipolice.gov.ae/wps/portal/home/opendata (accessed on 7 July 2023) https://www.dxbpp.gov.ae/ (accessed on 7 July 2023) |

Note: All variables are expressed in natural logarithms.

Table 2 and Figure 1 show some statistical characteristics and the evolution of time series paths from 1989 to 2021. The data, spanning more than three decades, enable any asymmetric behavior of the dependent variables to be captured, especially the impact of crime against a person on domestic investment.

**Table 2.** Descriptive statistics (1989–2021).

|  | **Inv** | **gdp** | **tra** | **crp** |
|---|---|---|---|---|
| Mean | 48,818.27 | 201,141.0 | 38,212.12 | 1772.848 |
| Median | 37,802 | 140,200.0 | 23,544.00 | 1620.000 |
| Maximum | 136,339 | 432,347.0 | 178,630.0 | 2973.000 |
| Minimum | 7942 | 31,764.00 | 7218.000 | 1198.000 |
| Std. Dev | 37,188.49 | 158,694.7 | 40,154.55 | 504.1769 |
| Skewness | 0.495676 | 0.187759 | 2.173619 | 0.838328 |
| Kurtosis | 2.173217 | 1.231313 | 7.133722 | 2.471302 |
| Jarque–Bera | 2.291227 | 4.495245 | 49.48094 | 4.249707 |
| Probability | 0.318029 | 0.105650 | 0.000000 | 0.119450 |
| Sum | 1,611,003 | 6,637,653 | 1,261,000 | 58,504.00 |
| Sum Sq. Dev. | 4.43 | $8.06 \times 10^{11}$ | $5.16 \times 10^{10}$ | 8,134,218 |
| Observations | 33 | 33 | 33 | 72 |

Table by the authors. Statistics were calculated using raw numerical values before applying logarithms.

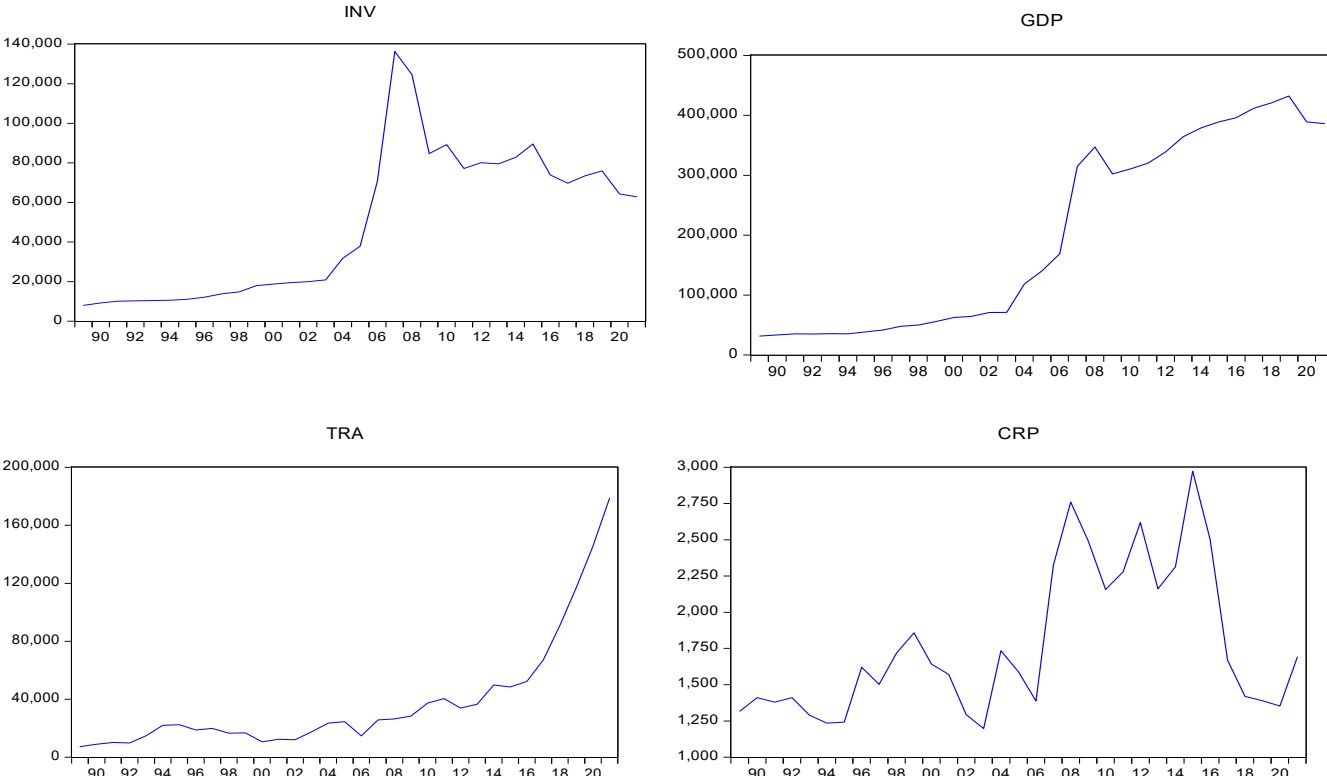

**Figure 1.** Variables data diagram. Figure by the authors.

There are many features in the descriptive statistical data presented in Table 2, the most noticeable of which is that the standard deviations for the variables compared with means indicate a variation in each variable, as shown in Figure 1. Also, the skewness values have positive skewness for all variables. Moreover, kurtosis values are less than 3, except for tra, which has high (leptokurtic) values. Also, variables skew towards abnormality, except for tra. This suggests that the model should include both a constant and a trend, so using logarithm-transformed data would be more accurate than using raw data. In addition, the nonlinear autoregressive distributed lag (NARDL) model would be more appropriate than utilizing the (Johansen and Juselius 1990) approach or ARDL.

Furthermore, Figure 1 depicts raw data showing an upward trend for domestic investment ($inv$) and ($gdp$) until March 2008 due to the global financial crisis, with little impact on the gross domestic product ($gdp$). Additionally, trade openness ($tra$) exhibits an upward trend, while crime against a person fluctuates over the study period. The figures indicate that using the logarithmic form of data would be more suitable than utilizing raw data, along with the recommended model incorporating constants and trends.

**4. Methodology**

Recent studies have applied several models to estimate crime impacts on economic growth, productivity, and investment, including ARDL and NARDL (Rauf et al. 2022; Goh and Law 2021; Adekoya and Razak 2017; Zeeshan et al. 2022; Habibullah and Baharom 2008; Ahmad et al. 2014). This study examined crime against a person by employing a novel nonlinear autoregressive distributed lag (NARDL) developed by Shin et al. (2014) as an extension of the widely recognized symmetric ARDL model introduced by Pesaran et al. (2001). NARDL has many advantages compared with the conventional cointegration tests by Engle and Granger (1987), (Johansen 1988) and Johansen and Juselius (1990) and ARDL, where ECM provides that the variables must be in the same order and not exceed I(2). ARDL overcomes the challenge of ECM by allowing the cointegration test to be executed between variables from different orders, stipulating that the integration of any variable does not exceed I(1) and the dependent variable is not stationary, i.e., I(0) (Adela 2023). However, it assumes that the relationship between the cointegrated variables in the long and short term is linear and lacks the ability to differentiate the asymmetric impacts of increasing or decreasing explanatory variables. NARDL can be extended for estimating short- and long-run nonlinearities through the positive and negative impacts of the independent variables (Allen and McAleer 2021) by dissecting explanatory variables into positive and negative partial sum squares. This approach incorporates the potential asymmetric impacts of positive and negative changes in explanatory variables on the dependent variable. Therefore, to explore the asymmetric impacts of crime against a person on domestic investment, we began by specifying the symmetric ARDL approach.

Consequently, the ARDL model can be expressed as follows:

$$invl_t = \alpha_0 + \alpha_1 gdpl_t + \alpha_2 tral_t + \alpha_3 crpl + \mu_t \qquad (1)$$

where

$invl$: domestic investment; $gdpl$: gross domestic product; $tral$: trade openness value; $crpl$: crime against a person; $\mu_t$: normal distributed error term; and $\alpha_0$, $\alpha_1$, $\alpha_2$, and $\alpha_3$: estimated coefficients. All variables were transformed into natural logarithms to assist with normalization and scaling. The practical assumption was that the effect of crime against a person may differ based on whether crime rates are decreasing or increasing. Therefore, the study posited the presence of an asymmetric relationship between crime against a person and domestic investment.

Thus, the original structure of the nonlinear ARDL model follows the conventional symmetric ARDL model presented by Pesaran et al. (2001), which supposes that the effects on the dependent variable are symmetrical for both short- and long-term relationships in response to increases or decreases in the explanatory variables. However, it recognizes that the dependent variable may exhibit greater sensitivity to downturns or upswings than the explanatory variable. The key characteristics of the ARDL model include the integration of variables at levels I(0) or I(1) or mutual cointegration, with the stipulation that integration levels do not exceed I(2). Additionally, the model assumes symmetric relationships among variables. Furthermore, it has the amplitude to combine both exogenous and endogenous variables, distinguishing it from the restricted vector autoregressive (VAR) model, which comprises solely endogenous variables.

In empirical applications, the influence of explanatory variables on the dependent variable fluctuates as it transitions from increases to decreases across various models.

Following the planning of descriptive data, the implementation of the nonlinear autoregressive distributed lag (NARDL) model entailed several steps. These steps included conducting unit root tests, selecting the appropriate lag length, estimating short- and long-term coefficients, conducting heteroscedasticity tests, and applying the Wald test to explore symmetries.

In this context, the ARDL model was derived from the unrestricted error correction model (ECM), which comprises identical lagged levels with unrestricted coefficients. The ECM model can be expressed by the following formula:

$$\Delta invl_t = \alpha_0 + \sum_{i=1}^{m} \alpha_1 \Delta invl_{t-i} + \sum_{j=0}^{n} \alpha_2 \Delta gdpl_{t-j} + \sum_{k=0}^{r} \alpha_3 \Delta tral_{t-k} + \sum_{l=0}^{q} \alpha_4 \Delta crpl_{t-l} + \beta_{z_{t-1}} + \mu_t \qquad (2)$$

ARDL substitutes the long-run term $\beta_1 invl_{t-1} + \beta_2 gdpl_{t-1} + \beta_3 tral_{t-1} + \beta_4 crpl_{t-1}$ instead of utilizing its residuals $ECT(\beta_{z_{t-1}})$.

The β coefficient corresponds to the error correction term, signifying the speed of adjustment towards long-term equilibrium. It is crucial for this coefficient to be negative and statistically significant to ensure convergence towards equilibrium. The absence of these conditions renders the model nonstationary.

The series of lagged residuals is presented in the following equation:

$$\beta_{z_{t-1}} = invl_{t-1} - b_0 - b_1 gdpl_{t-1} - b_2 tral_{t-1} - b_3 crpl_{t-1} \qquad (3)$$

Therefore, the initial formulation of the nonlinear ARDL model, which sustains the form of the conventional ARDL model introduced by Pesaran et al. (2001), is presented as the following equation:

$$\Delta invl_t = \alpha_0 + \sum_{i=1}^{m} \alpha_1 \Delta invl_{t-i} + \sum_{j=0}^{n} \alpha_2 \Delta gdpl_{t-j} + \sum_{k=0}^{r} \alpha_3 \Delta tral_{t-k} + \sum_{l=0}^{q} \alpha_4 \Delta crpl_{t-l}$$
$$+ \beta_1 invl_{t-1} + \beta_2 gdpl_{t-1} + \beta_3 tral_{t-1} + \beta_4 crpl_{t-1} + \mu_t \qquad (4)$$

When the short- run coefficients are $\alpha_1$, $\alpha_2$, $\alpha_3$, and $\alpha_4$, the ARDL long-run coefficients are $\beta_1, \beta_2$, $\beta_3$, and $\beta_4$, respectively.

The disturbance (white noise) term is $\mu_t$.

The NARDL methodology presented by Shin et al. (2014) encompasses an asymmetrical long-run regression. This involves dissecting a regressor into its positive and negative changes. In the context of the nonlinear autoregressive distributed lag (NARDL) model, it segregates the response of *invl* concerning the positive and negative fluctuations of crp*l*. Consequently, the model can be characterized as asymmetric as follows:

$$crpl_t = crpl_0 + crpl^- + crpl^+ \qquad (5)$$

where:

$$crp^+ = \sum_{j=1}^{t} \Delta crp_j^+ + \sum_{j=1}^{t} Max(\Delta crpl_j, \, 0) \qquad (6)$$

$$crpl^- = \sum_{j=1}^{t} \Delta crpl_j^- + \sum_{j=1}^{t} Min(\Delta crpl_j, 0) \qquad (7)$$

Consequently, the reformulation of Equation (1) should be as follows:

$$invl_t = \alpha_0 + \alpha_1 crpl_j^+ + \alpha_2 crpl_j^- + \alpha_3 gdpl_t + \alpha_4 tral_t + \mu_t \qquad (8)$$

$\mu_t$ represents the error correction.

Equation (5) facilitates the estimation of the asymmetrical impact on domestic investment resulting from an increase or decrease in the volume of crime against a person by replacing Equations (3) and (4) into Equation (5), resulting in the transformation of the model into the nonlinear autoregressive distributed lag (NARDL) model, as depicted below:

$$\Delta invl_t = \alpha_0 + \sum_{i=1}^{m} \alpha_{i1} \Delta dinvl_{t-1} + \sum_{i=0}^{n} \alpha_{i2}^+ \Delta crpl_{t-i}^+ + \sum_{i=0}^{t} \alpha_{i3}^- \Delta crpl_{t-i}^- + \sum_{i=0}^{r} \alpha_{i4} \Delta gdpl_{t-i} +$$
$$\sum_{i=0}^{q} \alpha_{i4} \Delta tral_{t-i} + \rho invl_{t-1} + \beta_1^+ crpl_{t-1}^+ + \beta_2^- crpl_{t-1}^- + \beta_3 gdpl_{t-1} + \beta_4 tral_{t-1} + \mu_t \qquad (9)$$

where $\alpha_1$, $\alpha_2$, $\alpha_3$, and $\alpha_{04}$ are NARDL short-run coefficients, and $\rho, \beta_2^+, \beta_3^-$, and $\beta_4$ are long-run coefficients with asymmetric terms associated with the cumulative sum of positive (negative) changes in $invl_t$.

As per F-statistics (Pesaran et al. 2001), $H_0$: $\rho_1 = \beta_2^+ = \beta_3^- = \beta_4 = 0$ and $H_1$: $\rho_1, \neq \beta_2^+ \neq \beta_3^- \neq \beta_3 = 0$. The long-run asymmetry is statistically significant when the null hypothesis that $\beta_2^+ = \beta_3^-$ is rejected.

Moreover, as indicated by the Wald test, the long-run asymmetric effect of $crpl_t$ on $invl_t$ is calculated by $\frac{-\beta_2^+}{\rho}, \frac{-\beta_3^+}{\rho}$, and the short-run asymmetric effect of $crpl_t$ on $invl_t$ is calculated by $\sum_{i=1}^n \alpha_{i2}^+, \sum_{i=1}^t \alpha_{i3}^-$, where, $H_0 : \frac{-\beta_2^+}{\rho} = \frac{-\beta_3^+}{\rho}$ and $H_1 : \frac{-\beta_2^+}{\rho} \neq \frac{-\beta_3^+}{\rho}$, Consequently, for the short run, $H_0 : \sum_{i=1}^n \alpha_{i2}^+, = \sum_{i=1}^t \alpha_{i3}^-$ and $H_1 : \sum_{i=1}^n \alpha_{i2}^+, \neq \sum_{i=1}^t \alpha_{i3}^-$.

Furthermore, asymmetric dynamic multipliers investigate the adjusted response of $invl_t$ to its new long-run nonlinearity equilibrium, which is influenced by prior positive or negative short-term dynamic shocks and the initial instabilities in $crpl_t$. The accumulative dynamic multiplier effects of $crpl^-$ and $crpl^+$ are estimated as follows:

$$m_k^+ = \sum_{i=0}^k \frac{\partial crpl_{t+i}}{\partial invl_i^+}, \ m_k^- = \sum_{i=0}^k \frac{\partial crpl_{t+i}}{\partial invl_i^-} \tag{10}$$

where $m_k^+$ and $m_k^-$ are long-run asymmetric coefficients, in which $k \to \infty$ and $m_k^+$ and $m_k^- \to \beta_2^+$ and $\beta_3^-$ respectively, and $m_h$ is attributed to the pivotal responsible for the volatilities. This illustrates how $crpl$ aligns with the pathways of NARDL in the long run based on a positive or negative change in $invl$.

## 5. Results and Discussion
### 5.1. Unit Root Tests

Table 3 displays the results of the unit root test for the estimated variables using both the augmented Dickey–Fuller (ADF) and Phillips–Perron (PP) tests. The findings revealed that the variables crpl and tral exhibited stationarity at $I(I)$ at the 1% significance level according to both ADF and PP. Also, gdpl and invl were stationary at level $I(I)$ at the 5% significance level for ADF and 10% significance level for PP. In light of these results, the vector autoregressive (VAR) and vector error correction model (VECM) proposed by Johansen and Juselius (1990) were considered for further analysis. Also, the autoregressive distributed lag (ARDL) model developed by Pesaran et al. (2001) was applied, where regressors are stationary at the same level, in order to investigate the existence of both short- and long-run cointegration among the variables and determine the coefficient representing the speed of adjustment in an error correction model. Both suppose that the impact of independent variables has a symmetric positive and negative impact on the dependent variable. Therefore, NARDL allows exogenous and endogenous variables to be decomposed into positive and negative partial sums of squares, assuming that the response of the dependent variable to increase or decrease depending on each independent variable is asymmetric.

The unit root test indicated that all variables were integrated at level $I(I)$.

**Table 3.** Unit root tests.

| | On Levels | | On First Differences | |
|---|---|---|---|---|
| | **ADF** | **PP** | **ADF** | **PP** |
| invl | −1.507247 | −0.789476 | −3.648211 ** | −3.332030 * |
| gdpl | −0.783830 | −4.238891 | −4.268683 ** | −4.238891 ** |
| crpl | −2.177064 | −5.544164 | −5.265264 *** | −5.5331634 *** |
| tral | −1.911051 | −3.562882 | −5.109150 *** | −6.290982 *** |

Table by the authors. Note: (*i*) The critical values for unit root tests at significance levels of 1%, 5%, and 10% were −4.28, −3.56, and 3.21 for both the augmented Dickey–Fuller (ADF) and Phillips–Perron (PP) tests. (ii) ***, **, and * denote statistical significance at 1%, 5%, and 10%, respectively. The lag lengths obtained from the VAR models extend to $p = 12$.

### 5.2. Selecting the Optimal Lag Lengths

Selecting the appropriate lag lengths is essential to avoid autocorrelated errors during the construction of the VAR model (Bruns and Stern 2019). The determination of lag lengths entails using the residuals of the VAR model to identify the lag without serial correlation problems. An alternative method for lag length selection was proposed by Johansen and Juselius (1990) utilizing five different criteria: Akaike information criterion (1974), LR, final prediction error, Hannan and Quinn (1979), and Yaniv and Schwartz (1991). However, these criteria have frequently yielded conflicting outcomes in the empirical financial literature (Othman et al. 2019).

The VAR model assumes various lag lengths up to a maximum lag of $p = 9$ until all the residuals exhibit no correlation (Table 4).

**Table 4.** Lag length.

| Lag | LogL | LR | FPE | AIC | SC |
|---|---|---|---|---|---|
| 0 | −35.38565 | NA | 0.000162 | 2.625710 | 2.812536 | 2.685477 |
| 1 | 78.91867 | 190.5072 * | $2.34 \times 10^{-7}$ * | −3.927911 * | −2.993780 * | −3.629075 * |
| 2 | 86.28791 | 10.31694 | $4.44 \times 10^{-7}$ | −3.352528 | −1.671091 | −2.814621 |
| 3 | 97.58032 | 12.79806 | $7.19 \times 10^{-7}$ | −3.038688 | −0.609946 | −2.261713 * |

Table by the authors. The results presented in Table 4 suggest that the optimal lag length is $p = 1$. Note: * indicates lag order selected by the criterion; LR: sequential modified LR test statistic (each test at the 5% significance level); FPE: final prediction error; AIC: Akaike information criterion; SC: Schwarz information criterion.

### 5.3. NARDL Estimation Results

The outcomes for both the unit roots test and the lag length selection are presented in Table 5 to illustrate the findings of the asymmetric bounds cointegration test introduced by Shin et al. (2014). This test was utilized to explore the presence of a long-term cointegration relationship among variables.

**Table 5.** Asymmetric bounds test.

| | Significance Level | Bounds | |
|---|---|---|---|
| | | Lower Bounds I(0) | Upper Bounds I(1) |
| F-statistic | 1% | 3.15 | 4.43 |
| 16.805396 | 5% | 2.45 | 3.61 |
| | 10% | 2.12 | 3.23 |
| T-statistic | 1% | −3.43 | −4.99 |
| −7.688449 | 5% | −2.86 | −4.38 |
| | 10% | −2.57 | −4.99 |

Table by the authors. The test was conducted on the equation with a constant based on the Pesaran et al. (2001) table. The null hypothesis being tested is that the long-run cointegrating relationship is symmetric.

The findings of the asymmetric bound test indicated the existence of an asymmetric long-run cointegrating relationship among variables at a significance level of 1%. This was evident by both the values of F = 16.805396 and T = −7.688449, which were above the upper bounds established at the 1% significance level. Furthermore, the adjustment of the long-run equilibrium appeared to be influenced by short-run fluctuations. However, the rate of adjustment towards equilibrium was 0.90% in each lag period.

The long-run asymmetric cointegrating equation is as follows:

$$\text{invl} = 9.0465 + 1.362\, gdpl^+ + 2.840 gdpl^- - 0.594 crpl^+ + 0.511 crpl^- + 0.195 tral^+ - 0.238 tral^-$$

$$(9.842732)\ *** \ (4.098493)\ *** \ (-2.885056)\ *** \ (3.448069)\ *** \ (1.845852)\ * \ (127.1821)\ *$$

Table 6 shows the long- and short-run estimation outcomes of the NARDL model. The results revealed the following:

1. The error correction term coefficient $ECT_{t-1}$ signified a negative long-run equilibrium at $-0.90$, indicating the speed of adjustment to correct the long-run equilibrium error during short-term fluctuations at the 1% significance level.

2. The relationship between domestic investment and crime against a person was characterized by an asymmetrical correlation, meaning that the effects on domestic investment vary depending on whether the crime against a person changes positively or negatively. Also, the effects of GDP and trade openness on crime against a person exhibited asymmetrical patterns.

3. The $crpl^+$ long-run asymmetric coefficients demonstrated that crime against a person has a significantly negative asymmetric impact on domestic investment in Dubai at the 1% significance level. Hence, a 1% increase in crime against a person ($crpl^+$) led to a $-0.6\%$ decrease in domestic investment, whereas the negative long-run asymmetric coefficient ($crpl^-$) was 0.5. Consequently, a 1% reduction in crime against a person resulted in an increase in domestic investment of 0.5. This suggests that crime against individuals in Dubai can negatively impact domestic investment due to safety concerns. Stability and safety are crucial for investors, and a rise in violent crimes can create an atmosphere of insecurity, discouraging both individual and institutional investors. Due to Dubai's heavy reliance on tourism, any surge in such crimes can tarnish its reputation, resulting in negative perceptions among potential investors. In turn, this can lead to a decline in tourism revenue, a cascading effect on other industries, and a subsequent erosion of investor confidence. Also, investors highly value legal stability and investment predictability. Uncertainty stemming from concerns about crime and its implications for regulations can deter investments. Therefore, Dubai Police applies cutting-edge techniques for speedily detecting crimes, especially homicide. Moreover, the Emirates Police departments strive to compete with each other for swift detection of crimes against a person, disseminating this information via websites and newspapers. Additionally, a secure environment is pivotal for attracting and retaining skilled professionals. Growing crime rates may discourage talented individuals from choosing Dubai as a place to reside and work, potentially leading to a labor shortage. Moreover, rising crime inhibits consumer confidence, affecting spending habits and business revenues. Furthermore, elevated crime rates can result in increased operating costs related to security and safety measures, which directly impact profitability. As a result of perceived risks, investors may request higher returns, impacting the cost of capital for businesses.

4. The positive ($gdpl^+$) and negative ($gdpl^-$) long-run asymmetric coefficients were 1.4% and 2.8%, respectively, at the 1% significance level. This indicates that changes in GDP have an asymmetric impact on domestic investment, suggesting that both positive and negative changes in GDP positively influence the volume of domestic investment. Hence, a 1% increase in GDP led to a 1.4% increase in domestic investment, which aligns with the observed growth rates in both domestic investment and GDP spanning the study period. Meanwhile, a decrease in GDP led to an increase in domestic investment of 2.8%. Notably, the negative coefficient exceeded the positive coefficient, indicating that an increase in the GDP motivates the public and private sectors to enhance domestic investment as well as stimulates the government to intensify its expenditure to elevate the economy. Meanwhile, a decrease in GDP prompts the government to compensate for the decrease in private investment in a strategic plan, which leads to public investment exceeding the decline in private investment. However, it is important to note that short-term effects were not observable because the lag period was one year. Therefore, the increase in exploratory variables would affect the dependent variable ($invl^+$) one year later.

**Table 6.** Estimates of asymmetric coefficients.

| Variable | Long-Run Asymmetric Cointegration Coefficient Equation. | | | |
|---|---|---|---|---|
| | Coefficient | Std. Error | t-Statistic | Prob. |
| $gdpl^{+}$ | 1.222052 *** | 0.138349 | 9.842732 | 0.0000 |
| $gdpl^{-}$ | 2.547997 *** | 0.692750 | 4.098493 | 0.0000 |
| $crpl^{+}$ | −0.594074 *** | 0.205914 | −2.885056 | 0.0073 |
| $crpl^{-}$ | 0.510747 *** | 0.148126 | 3.448069 | 0.0051 |
| $tral^{+}$ | 0.174796 * | 0.105520 | 1.845852 | 0.0724 |
| $tral^{-}$ | −0.213581 * | 0.131290 | −1.812722 | 0.0866 |
| C | 8.118518 *** | 3.204524 | 6.555294 | 0.0000 |
| $ECT_{t-1}$ | −0.897426 *** | 0.074007 | −12.12630 | 0.0000 |
| $R^2$ | 0.830554 * | | | 0.064630 |

Table by the authors. Note: (+) and (–) denote the positive and negative partial sums, while *** and * indicate statistical significance at the 1% and 10% levels, respectively, *p*-values are included in square brackets. *WLR* and *WSR* correspond to the Wald test for asymmetry in the long run and short run, respectively. The null hypothesis assumes coefficients. *WSR* and *WLR* denote the short- and long-run Wald statistic symmetries, respectively.

Furthermore, the long-term coefficients pertaining to trade openness revealed an asymmetric relationship between trade openness and domestic investment, showing a positive relationship in both $tral^{+}$ and $tral^{-}$. The Dubai economy has a strong relationship with the world economy in both economic upward and downward trends, which is reflected in economic growth and domestic investment. In addition, there are no currency restrictions in Dubai as investors can hold fully convertible accounts in dirhams or foreign currency (ITA, United Arab Emirates 2023). Dubai is the center of a trade hub in the Southern Hemisphere. From 2020 to 2023, it was the fifth-largest transshipment center in the world (Xinhua-Baltic International Shipping Centre, Development Index Report 2023), behind Singapore, London, Shanghai, and Hong Kong, making it a desirable place for numerous businesses to open branches.

Table 7 presents the results of the Wald test, showing we can reject the null hypothesis. This rejection indicates significant asymmetry in both the short and long term, suggesting that the changes in these intervals differ from each other and confirming the presence of asymmetric effects. Therefore, it can be inferred that crimes against a person has an impact on domestic investment with varying positive and negative magnitudes.

**Table 7.** Test for asymmetries.

| Wald Statistic Test | $\chi^2$ Statistic |
|---|---|
| WLR | 20.46357 *** (0.000) |

Table by the authors. Note. The null hypothesis is that the coefficients are symmetric. *WLR* denotes the long-run Wald statistic asymmetries. *** denote statistical significance at the 1% levels, respectively. $\chi^2$ is chi-square. The *p*-value is represented in parentheses. $W_{LR}$ is $\frac{-\beta_2^{+}}{\rho} = \frac{-\beta_3^{+}}{\rho}$.

Furthermore, Figure 2 illustrates the results of the CUSUM and CUSUMSQ statistics stability tests (Brown et al. 1975) to determine trajectory significance at the 95% confidence bounds. It indicates the rejection of the null hypothesis. Consequently, we infer that all parameters in this regression are stable.

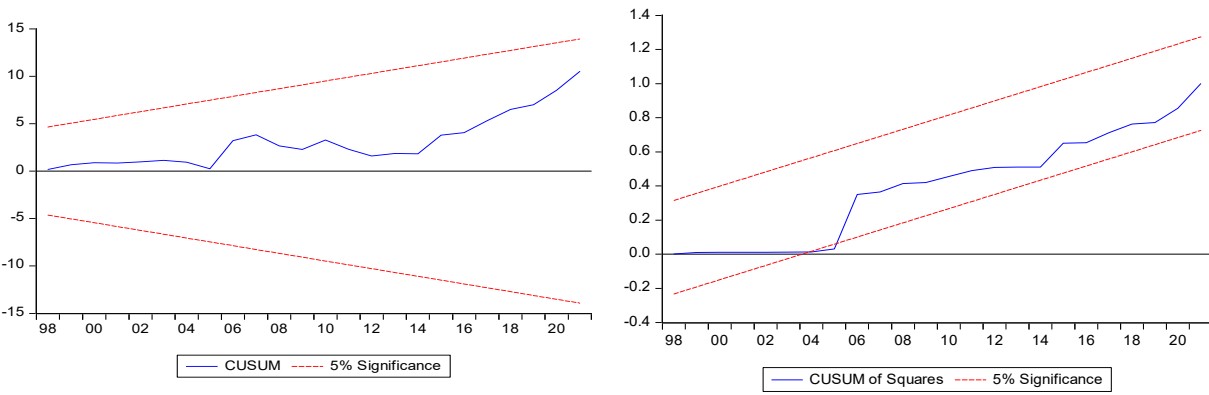

**Figure 2.** CUSUM and CUSUMQ tests for model stability. Figure by the author.

## 6. Conclusions

This study investigated the asymmetric impacts of a particular type of crime against a person on domestic investment, focusing on Dubai as an emerging megacity. The asymmetric long-term relationship between crime against individuals and domestic investment in Dubai was revealed utilizing a nonlinear autoregressive distributed lag (NARDL) model to examine the positive and negative crime changes against individuals on domestic investment. This study provides a clear insight into whether crimes against a person could pose a potential threat to domestic investment in the foreseeable future.

The primary findings revealed a robust and statistically significant long-term equilibrium relationship, highlighted by a −0.90 coefficient for the speed of adjustment of the error correction term. This relationship was observed between positive and negative partial sum squares of crime against a person, as confirmed by the asymmetric bounds test developed by Shin et al. (2014). Additionally, changes in crime against individuals demonstrated an asymmetric impact on domestic investment, as confirmed by the Wald test. Long-run estimations of coefficients using the nonlinear autoregressive distributed lag (NARDL) model indicated a negative effect of crime against a person on domestic investment in Dubai. Notably, the partial sum squares of long-run asymmetric coefficients of crime against a person exhibited positive and negative changes of −0.6% and 0.5%, respectively. These values indicate an inverse proximity effect regarding the impact on domestic investment. The results are consistent with many recent studies concerning the relationship between crime and economic variables (Garriga and Phillips 2022; Sugiharti et al. 2022; Simo-Kengne and Bitterhout 2023; Raj and Kalluru 2023; Boamah et al. 2023; Yu et al. 2023; Magableh et al. 2023; Forgione and Migliardo 2023), suggesting that the Dubai Police needs to continue conducing activities in order to minimize the impact of crime against a person on domestic investment and economic activity. This management involves the utilization of advanced technology and artificial intelligence. Additionally, the results indicate that investors prioritize security and stability when making investments, and a rise in violent crimes can potentially foster an atmosphere of insecurity, which deters both individual and institutional investors.

Dubai's economy heavily relies on tourism. Therefore, any surge in such criminal incidents can tarnish its reputation, resulting in unfavorable impressions among potential investors. As a result, this could diminish tourism revenue and have cascading impacts on other industries, ultimately affecting investor confidence. Moreover, a safe environment plays a significant role in attracting and retaining skilled specialists, causing a reduction in the employment rate. However, an increase in crime against a person may dissuade talented people from choosing Dubai as their home for residence and workplace, leading to a shortage of skilled professionals. Furthermore, rising crime against a person undermines consumer confidence, influencing spending behaviors and corporate revenues, which can affect the cost of production and investors' profitability, as well as necessitating an increase in public expenditure on security and safety measures.

Therefore, this study suggests the imperative for the Dubai Police to sustain their efforts in diminishing crime against a person by implementing various programs employing artificial intelligence and cutting-edge technology to enhance public safety and security. The results can be applied to other cities in the Emirates, particularly Abu Dhabi. Additionally, the study emphasizes the importance of enhanced collaboration among all security policymakers in the Emirates towards achieving a 0% crime rate against a person. Moreover, the study highlights the need for further studies to investigate the effect of crime on domestic investment and economic growth in the UAE compared to other emerging economies. It also suggests examining the relationship between variables that may have a direct relationship with crime, such as financial variables and the creation or closure of companies. Furthermore, it is recommended that an economic risk measurement variable be developed to gauge the risk generated by crime.

Additionally, the long-run asymmetric coefficients suggest that both positive ($gdpl^+$) and negative ($gdpl^-$) changes in GDP have an asymmetric effect on domestic investment. Moreover, both revealed a positive influence. Remarkably, the negative coefficient surpassed the positive coefficient, indicating that an increase in GDP encourages both the private and public sectors to boost their investment. Furthermore, a decline in GDP spurs the government to increase its economic and social expenditures to offset the reduction in private investment, leading to public investment exceeding the decrease in private investment.

The long-term coefficients pertaining to trade openness reveal an asymmetric relationship between trade openness and domestic investment, showing a positive relationship in both ($tral^+$, $tral^-$). The Dubai economy has a significant relationship with the world economy in both economic upward and downward trends, which is reflected in economic growth and domestic investment. Additionally, Dubai is the center of a trade hub in the Southern Hemisphere. It continues to rank first in the Arab region and was the world's fifth-largest transshipment center from 2020 to 2023, behind Singapore, London, Shanghai, and Hong Kong, making Dubai an attractive destination for numerous businesses to establish branches (Xinhua-Baltic International Shipping Centre, Development Index Report 2023).

**Author Contributions:** Conceptualization, H.A.; methodology, H.A.; software, H.A.; validation, H.A., W.A.; formal analysis, H.A., W.A.; investigation, H.A.; resources, H.A., W.A.; data curation, H.A., W.A.; writing—original draft preparation, H.A.; writing—review and editing, H.A., W.A.; visualization, H.A.; supervision, H.A.; project administration, H.A.; funding acquisition, H.A., W.A. All authors have read and agreed to the published version of the manuscript.

**Funding:** This paper received no external funding.

**Data Availability Statement:** The data sources are presented in Table 1.

**Conflicts of Interest:** The authors declare no conflicts of interest.

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
