# Peer review of "The Impact of Crime against a Person on Domestic Investment in Dubai"

_jrfm, doi:10.3390/jrfm17020081_

Round 1

Reviewer 1 Report

Comments and Suggestions for Authors

No doubt the paper quality is excellent and the topic of study is also very relevant. The only thing I suggest the authors to throw some light on the problems faced by migrant workers which is undoubtedly a type of crime. This means that still to this day, migrant workers in the UAE face severe exploitation that restricts their freedom to change employers, receive their wages, and leave the country. Additionally, the migrant workers' countries of origin fail to protect their citizens sufficiently. Although the UAE's labour law explicitly prohibits and criminalizes all forms of forced labour, rampant labour abuses including indebtedness, passport confiscation and job immobility leave migrant workers in highly vulnerable situations that in many cases even amount to forced labour.

Author Response

Comments and Suggestions for Authors

No doubt the paper quality is excellent, and the topic of study is also very relevant. The only thing I suggest is the authors to throw some light on the problems faced by migrant workers which is undoubtedly a type of crime. This means that still to this day, migrant workers in the UAE face severe exploitation that restricts their freedom to change employers, receive their wages, and leave the country. Additionally, the migrant workers' countries of origin fail to protect their citizens sufficiently. Although the UAE's labour law explicitly prohibits and criminalizes all forms of forced labour, rampant labour abuses including indebtedness, passport confiscation and job immobility leave migrant workers in highly vulnerable situations that in many cases even amount to forced labour.

Dear respected Professor,

Many thanks for reviewing my paper.

The revision:

Your suggestion related to the problems faced by migrant workers is not within the scope of the paper, which focuses on the impact of a specific type of crime, namely the crime of Homicide, on domestic investment in Dubai. It can be identified in the paper if workers facing injustice are driven to commit Homicide. However, the labor laws here favor workers more than employers, and any complaint from the worker is investigated and addressed. This happened with my friend. It is also easy to transition from one job to another; in case of employer disagree, the worker can resort to the Ministry of Labor, which grants permission to change jobs without the employer's permission. Additionally, withholding passports from the workers is prohibited.

Kindly find the attached labor laws in the United Arab Emirates, particularly, the Federal Decree-Law No. (33) of 2021

https://www.mohre.gov.ae/en/laws-and-regulations/laws.aspx

Much appreciated Professor

Reviewer 2 Report

Comments and Suggestions for Authors

I found the topic of the work very interesting. However, there are a number of issues that should be clarified:

      -  In the introduction it should be made clear how the paper is structured, the different headings that integrate it.

-       -  It would be interesting to include or refer to some variables that may have a direct relationship with crime, such as the creation or closure of companies.

-        -  It would be interesting to define a variable to measure the risk generated by crime.

-         - In the literature review you talk about opportunity cost. What is the opportunity cost generated by crime in Dubai?.

Congratulations

Author Response

Comments and Suggestions for Authors

I found the topic of the work very interesting. However, there are a number of issues that should be clarified:

- In the introduction it should be made clear how the paper is structured, the different headings that integrate it.

The statement has been added:

The study is structured into five sections as bellows: the first is the introduction, followed by the literature review. The third section describes the data and descriptive analysis. The fourth section presents the methodology. The fifth section exhibits the empirical results and discussion, Finally, the concluding section encompasses conclusions and policy implications. Thank you

-  It would be interesting to include or refer to some variables that may have a direct relationship with crime, such as the creation or closure of companies.

Thank you for your recommendation. The study investigates the impact of a specific type of crime on domestic investment. Therefore, adding variables to study to examine the impact on crime requires separate study to analyze from a different direction, as the dependent variable is domestic investment. Therefore, investigating the impact of business activities and financial variables on crime necessitates separate study. I will add your valuable recommendation to the conclusion applications. Your recommendation will encourage researchers to conduct further studies on the impact of business and corporate variables on crime.

- It would be interesting to define a variable to measure the risk generated by crime.

Thank you for the suggestion, I considered the dependent variables as a variable to measure the risk generated by crime, introducing additional variables related to measuring crime risk may divert our attention from this primary objective. Therefore, the study focuses on the dependent variable as an economic risk measurement for maintaining clarity and extracting the impact on domestic investment. I also added this valuable suggestions to the recommendation to enhance further studies for these significant aspects.

- In the literature review you talk about opportunity cost. What is the opportunity cost generated by crime in Dubai?

I referred to the opportunity cost in the literature review for the criminal on the microeconomic level. Which includes sacrificing legitimate employment but also concealed and unpredictable expenses stemming from potential imprisonment, societal dislocation, and the psychological strain of engaging in illicit acts. I add for the society. 

The statement has been added.

Moreover, The opportunity cost for Dubai economy represents the direct costs stemming from the loss of production diversionnt, such as cases of homicide resulting a decline in gross domestic product, the costs of preventing crime and criminals’ rehabilitation, the costs of restrictions on human performance, the divert of economic resources, and economic activity. Also, indirect costs, such as negative impacts on the gross domestic products due to multiplier effects, the cost of medical care, diminishing investment in human capital, lower domestic and foreign investment due to uncertainty and investment costs, and the tendency of small businesses to move towards the informal economy. Therefore, the opportunity cost for society can be calculated by the difference between the level of GDP in the absence of crime and the value of GDP in the event of crime occurring.

Thank you, Dear Professor- Much appreciated.

Reviewer 3 Report

Comments and Suggestions for Authors

Dear Authors

Please review as follow

1- Have a finding section, set clearly what you have found

3- Where is discussion? you need to discuss your results compared to the other similar studies

3- Conclusions are not findings, take the findings out

4- Have a policy recommendation section

5- Follow the journal guide for format as it seems out of format 

Comments on the Quality of English Language

Reviewed is required 

Author Response

Comments and Suggestions for Authors

Dear Authors

 Please review as follow

1- Have a finding section, set clearly what you have found

I hope I revised it to be more clearly. Thank you.

3- Where is the discussion? you need to discuss your results compared to the other similar studies

I have added a comparison with the following studies:

These results are consistent with many recent studies concerning the relationship between crime and economic variables (Garriga, et al. 2022; Sugiharti et al. 2022; Simo-Kengne. and Bitterhout 2023; Raj and Kalluru., 2023; Boamah et al., 2023; Yu et al., 2023; Magableh et al., 2023; Forgione and Migliardo., 2023). The majority of studies investigate the effects of corruption or crime in general rather than crime against individuals. Notably, this topic is not extensively explored in existing literature, potentially due to the difficulty of obtaining the data.

3- Conclusions are not findings, take the findings out

I revised it to be clearer. Much appreciated.

4- Have a policy recommendation section

I have revised the recommendations. Thank you.

(Therefore, the study suggests the imperative for the Dubai Police to sustain their efforts in diminishing crime against a person by implementing various programs employing artificial intelligence and cutting-edge technology to enhance public safety and security. The results can be applied to other Emirates, particularly Abu Dhabi. Additionally, the study emphasizes the importance of security policymakers enhancing collaboration among all Emirates towards achieving a 0% crime rate against a person. Furthermore, the study urges further studies to investigate the effect of crime on domestic investment and economic growth in the UAE compared to other emerging economies. It also suggests examining the relationship between variables that may have a direct relationship with crime, such as financial variables and the creation or closure of companies. In addition, define an economic risk measurement variable to gauge the risk generated by crime.)

5- Follow the journal guide for format as it seems out of format

I have checked the paper and followed your advice. Many thanks.

Comments on the Quality of English Language

Review is required.

I proofread further; Thank you.

Much appreciated Professor

Reviewer 4 Report

Comments and Suggestions for Authors

The paper titled "The Impact of Crime Against a Person on Domestic Investment in Dubai" investigates the influence of crime on domestic investment in Dubai from 1989 to 2021. The study employs a novel nonlinear autoregressive distributive lag (NARDL) approach to analyze the asymmetric impact of crime on domestic investment. It contributes to the understanding of the economic implications of crime in emerging economies, particularly in the context of a global city like Dubai.

The research topic is original, addressing the specific impact of crime on domestic investment in Dubai, an area not extensively covered in existing literature. The use of NARDL to analyze asymmetric impacts offers an advancement in the methodology for such studies. The study aligns with the journal's aim of contributing to the understanding of economic phenomena. It fits well within the scope of economic analysis, particularly in the context of emerging economies and their global cities. The findings are significant, demonstrating an asymmetric impact of crime on domestic investment. The study interprets these results appropriately, offering insights into the economic consequences of crime in Dubai. The conclusions are justified and supported by robust empirical analysis. The article is well-written, presenting data and analyses clearly and coherently. The use of NARDL and the comprehensive data set from 1989 to 2021 demonstrate high standards in research methodology and presentation. The study is technically sound, with a well-designed methodology. The use of NARDL is appropriate, and the data is robust enough to support the conclusions drawn. The methods, tools, and analyses are described in detail, allowing for reproducibility of results. The topic is of interest to a wide readership, particularly those interested in the economic impacts of social issues like crime, as well as those focusing on emerging economies and global cities. The paper offers some benefits to the field, providing insights into an under-researched area. It advances current knowledge and addresses important economic questions using sound methodology. However, the current version of the manuscript needs to be improved to publish in a high-quality journal. My comments to improve the manuscript are given below:

1)      While the use of NARDL is innovative, the manuscript could benefit from a more detailed explanation of this methodology, particularly for readers who may not be familiar with it. This could include a clearer rationale for choosing NARDL over other models.

2)      Analysis: The paper might be strengthened by including comparative analysis with other major cities that have faced similar issues. This would provide a broader context and enhance the understanding of Dubai's unique situation.

3)      The study focuses primarily on crime, but it could also consider other economic factors that might influence domestic investment in Dubai, such as political stability, economic policies, or global economic trends.

4)      Acknowledging any limitations in the data used and suggesting areas for future research could provide a more balanced view and encourage further study in this field. Also, expanding the discussion on policy implications based on the findings would be beneficial. This could include recommendations for policymakers in Dubai and similar economies.

5)      Incorporating more graphical representations of data could enhance the readability and appeal of the paper, making the findings more accessible to a broader audience. In addition,  Including case studies or specific examples of how crime has directly affected investment in certain sectors or areas in Dubai would add depth to the paper.

Minor comments:

1)      A more comprehensive review of existing literature, especially studies on crime and investment in other emerging economies, could provide a richer theoretical framework.

2)      The conclusion could be strengthened by summarizing the main findings more succinctly and highlighting the study's contribution to the existing body of knowledge more clearly.

3)      A thorough editing and proofreading pass could help to eliminate any remaining grammatical or typographical errors, ensuring the paper's professionalism.

Comments on the Quality of English Language

Minor editing of English language required

Author Response

Comments and Suggestions for Authors

The paper titled "The Impact of Crime Against a Person on Domestic Investment in Dubai" investigates the influence of crime on domestic investment in Dubai from 1989 to 2021. The study employs a novel nonlinear autoregressive distributive lag (NARDL) approach to analyze the asymmetric impact of crime on domestic investment. It contributes to the understanding of the economic implications of crime in emerging economies, particularly in the context of a global city like Dubai.

The research topic is original, addressing the specific impact of crime on domestic investment in Dubai, an area not extensively covered in existing literature. The use of NARDL to analyze asymmetric impacts offers an advancement in the methodology for such studies. The study aligns with the journal's aim of contributing to the understanding of economic phenomena. It fits well within the scope of economic analysis, particularly in the context of emerging economies and their global cities. The findings are significant, demonstrating an asymmetric impact of crime on domestic investment. The study interprets these results appropriately, offering insights into the economic consequences of crime in Dubai. The conclusions are justified and supported by robust empirical analysis. The article is well-written, presenting data and analyses clearly and coherently. The use of NARDL and the comprehensive data set from 1989 to 2021 demonstrate high standards in research methodology and presentation. The study is technically sound, with a well-designed methodology. The use of NARDL is appropriate, and the data is robust enough to support the conclusions drawn. The methods, tools, and analyses are described in detail, allowing for reproducibility of results. The topic is of interest to a wide readership, particularly those interested in the economic impacts of social issues like crime, as well as those focusing on emerging economies and global cities. The paper offers some benefits to the field, providing insights into an under-researched area. It advances current knowledge and addresses important economic questions using sound methodology. However, the current version of the manuscript needs to be improved to publish in a high-quality journal. My comments to improve the manuscript are given below:

1) While the use of NARDL is innovative, the manuscript could benefit from a more detailed explanation of this methodology, particularly for readers who may not be familiar with it. This could include a clearer rationale for choosing NARDL over other models.

I updated the text to make it more clear why NARDL was selected over alternative models. Many thanks.

(Recent studies have applied several models in estimating crime impacts on economic growth, productivity, and investment including ARDL and NARDL (Rauf et al., 2022; Goh et al., 2021; Adekoya et al., 2017; Zeeshan et al., 2022; Habibullah et al., 2008; Ahmed et al., 2014). This paper focuses on crime against a person, employing a novel nonlinear Autoregressive distributes lag (NARDL) developed by Shin et al. (2014), as an extension of the widely recognized symmetric ARDL model introduced by Pesaran et al. (2001). The NARDL has many advantages compared with the conventional cointegration test by Engle & Granger (1987), Johnson (1988) and Johnson & Juselius (1990), and the ARDL. where ECM provides that the variables must be in the same order and not exceed I(2). Despite the ARDL overcomes the challenge of ECM by allowing execute the cointegration test between variables form different orders, stipulating that the integration of any variable does not exceed I(1), and the dependent variable is not stationary I(0) (Adela, 2023). However, it assumes that the relationship between the cointegrated variables in the long- and short-term is linear and lacks the ability to differentiate the asymmetric impacts of increasing or decreasing explanatory variables. The NARDL extended for estimating the short- and long-run nonlinearities through positive and negative impacts of the independent variables (Allen and McAleer, 2021), by dissecting explanatory variables into positive and negative partial sum squares, this approach incorporates potential asymmetric impacts of positive and negative changes in explanatory variables on the dependent variable. Therefore, to explore the asymmetric impacts of crime against a person on domestic investment, we begin by specifying the symmetric ARDL approach.)

2) Analysis: The paper might be strengthened by including comparative analysis with other major cities that have faced similar issues. This would provide a broader context and enhance the understanding of Dubai's unique situation.

I have added a comparison with the following studies:

These results are consistent with many recent studies concerning the relationship between crime and economic variables (Garriga, et al. 2022; Sugiharti et al. 2022; Simo-Kengne. and Bitterhout 2023; Raj and Kalluru., 2023; Boamah et al., 2023; Yu et al., 2023; Magableh et al., 2023; Forgione and Migliardo., 2023). The majority of studies investigate the effects of corruption or crime in general rather than crime against individuals. Notably, this topic is not extensively explored in existing literature, potentially due to the difficulty of obtaining the data.

3) The study focuses primarily on crime, but it could also consider other economic factors that might influence domestic investment in Dubai, such as political stability, economic policies, or global economic trends.

The research comprised primary explanatory variables besides crime, such as gross domestic product and trade openness, because the economy in Dubai is coherent to the global economic trend. But dummy variables such as economic stability or economic policies are not included due to there are no fluctuations in either of them during the study period that could affect the results. However, crime is the primary independent variable that the study examines the influence on domestic investment.

4) Acknowledging any limitations in the data used and suggesting areas for future research could provide a more balanced view and encourage further study in this field. Also, expanding the discussion on policy implications based on the findings would be beneficial. This could include recommendations for policymakers in Dubai and similar economies.

Therefore, the study suggests the imperative for the Dubai Police to sustain their efforts in diminishing crime against a person by implementing various programs employing artificial intelligence and cutting-edge technology to enhance public safety and security. The results can be applied to other Emirates, particularly Abu Dhabi. Additionally, the study emphasizes the importance of security policymakers enhancing collaboration among all Emirates towards achieving a 0% crime rate against a person. Moreover, the study highlights the need for further studies to investigate the effect of crime on domestic investment and economic growth in the UAE compared to other emerging economies. It also suggests examining the relationship between variables that may have a direct relationship with crime, such as financial variables and the creation or closure of companies. Furthermore, there is a recommendation to develop an economic risk measurement variable to gauge the risk generated by crime.

5) Incorporating more graphical representations of data could enhance the readability and appeal of the paper, making the findings more accessible to a broader audience. In addition, Including case studies or specific examples of how crime has directly affected investment in certain sectors or areas in Dubai would add depth to the paper.

 It is difficult to obtain such information.

Minor comments:

1)  A more comprehensive review of existing literature, especially studies on crime and investment in other emerging economies, could provide a richer theoretical framework.

I have amended the literature. It incorporates studies related to the relationship between crime and investment as well as economic growth across various emerging economies such as Nigeria, India, Romania, Mexico, ECOWAS region, and Malysia.

2) The conclusion could be strengthened by summarizing the main findings more succinctly and highlighting the study's contribution to the existing body of knowledge more clearly.

I revised it. But I couldn't make it more succinctly.

3) A thorough editing and proofreading pass could help to eliminate any remaining grammatical or typographical errors, ensuring the paper's professionalism.

I proofread further; much appreciated.

Comments on the Quality of English Language

Minor editing of English language required.

I have checked the editing of the English language.

Much appreciated Dear Professor

Reviewer 5 Report

Comments and Suggestions for Authors

Very interesting paper - suggest that the following points be addressed to make the paper better.

1. It should be clarified as to what kind of investment is being talked about - the import of the paper is of external investment, but that needs to be clearly stated. 

2. While the incidence of crime is taken in to consideration, the efficacy and efficiency (in terms of timely and visible action) of redressal in terms of legal provisions is not discussed. The same should be a part of the discussions as that also helps build investor confidence and hence influences investment decisions.

3. Ease of withdrawal of investments should also be considered in determining the factors influencing investments, particularly for external investments.

4. The data spans 22 years and has seen many major changes in the law enforcement segment - it might be relevant to look at data in different regimes separately as they can otherwise be confounded effects.

5. The use of NARDL needs to be better justified as the methodological choice. Whay other alternates existed and why is this approach the best suited one?

6. Finally, which are the industries being considered and how generic would the findings be to other industries? It might be a good idea to classify the effect in terms of industries which are easy to relocate and thus have an immediate impact but can also be easily reversed (like tourism) or industries which are more stable and difficult to relocate (like port infrastructure) but if withdrawn, rarely it will come back.

Comments on the Quality of English Language

Generally, the english is fine but the paper can benefit for professional copy editing.

Author Response

Comments and Suggestions for Authors

Very interesting paper - suggest that the following points be addressed to make the paper better.

  1. It should be clarified as to what kind of investment is being talked about - the import of the paper is of external investment, but that needs to be clearly stated.

The paper investigates domestic investment, without differentiating between external and internal investment.

  1. While the incidence of crime is taken into consideration, the efficacy and efficiency (in terms of timely and visible action) of redressal in terms of legal provisions is not discussed. The same should be a part of the discussions as that also helps build investor confidence and hence influences investment decisions.

I have added more explanation when discussing the NARDL estimation results “Therefore, Dubai Police applies cutting-edge techniques for speedily detecting crimes, especially homicide. Moreover, the Emirates Police departments strive to compete with each other for swift detection of crime against a person, disseminating this information via websites and newspapers.”

  1. Ease of withdrawal of investments should also be considered in determining the factors influencing investments, particularly for external investments.

I have added more explanation when discussing the NARDL estimation results.

(In addition, There are no currency restrictions in Dubai, as investors can hold fully convertible accounts in in dirhams or foreign currency (ITA, United Arab Emirates, Country Commercial Guide, 2023).4. The data spans 22 years and has seen many major changes in the law enforcement segment - it might be relevant to look at data in different regimes separately as they can otherwise be confounded effects). Much grateful.

  1. The use of NARDL needs to be better justified as the methodological choice. Whay other alternates existed and why is this approach the best suited one?

I updated the text to make it more clear why NARDL was selected over alternative models. Many thanks.

(Recent studies have applied several models in estimating crime impacts on economic growth, productivity, and investment including ARDL and NARDL (Rauf et al., 2022; Goh et al., 2021; Adekoya et al., 2017; Zeeshan et al., 2022; Habibullah et al., 2008; Ahmed et al., 2014). This paper focuses on crime against a person, employing a novel nonlinear Autoregressive distributes lag (NARDL) developed by Shin et al. (2014), as an extension of the widely recognized symmetric ARDL model introduced by Pesaran et al. (2001). The NARDL has many advantages compared with the conventional cointegration test by Engle & Granger (1987), Johnson (1988) and Johnson & Juselius (1990), and the ARDL. where ECM provides that the variables must be in the same order and not exceed I(2). Despite the ARDL overcomes the challenge of ECM by allowing execute the cointegration test between variables form different orders, stipulating that the integration of any variable does not exceed I(1), and the dependent variable is not stationary I(0) (Adela, 2023). However, it assumes that the relationship between the cointegrated variables in the long- and short-term is linear and lacks the ability to differentiate the asymmetric impacts of increasing or decreasing explanatory variables. The NARDL extended for estimating the short- and long-run nonlinearities through positive and negative impacts of the independent variables (Allen and McAleer, 2021), by dissecting explanatory variables into positive and negative partial sum squares, this approach incorporates potential asymmetric impacts of positive and negative changes in explanatory variables on the dependent variable. Therefore, to explore the asymmetric impacts of crime against a person on domestic investment, we begin by specifying the symmetric ARDL approach.)

  1. Finally, which are the industries being considered and how generic would the findings be to other industries? It might be a good idea to classify the effect in terms of industries which are easy to relocate and thus have an immediate impact but can also be easily reversed (like tourism) or industries which are more stable and difficult to relocate (like port infrastructure) but if withdrawn, rarely it will come back.

There is a lack of information on crime related to specific industries to investigate its effectiveness.

Comments on the Quality of English Language

Generally, the English is fine, but the paper can benefit for professional copy editing.

I have checked the editing of the English language.

Much appreciated Dear Professor.

Round 2

Reviewer 5 Report

Comments and Suggestions for Authors

The responses are fine.